# Social Cognition Impairments in Association to Clinical, Cognitive, Mood, and Fatigue Features in Multiple Sclerosis: A Study Protocol

**Triantafyllos K. Doskas** [1,2], **Foteini Christidi** [2], **Kanellos C. Spiliopoulos** [1,3], **Dimitrios Tsiptsios** [2,*], **George D. Vavougios** [4], **Anna Tsiakiri** [2], **Theofanis Vorvolakos** [5], **Christos Kokkotis** [6], **Ioannis Iliopoulos** [2], **Nikolaos Aggelousis** [6] and **Konstantinos Vadikolias** [2]

1   Neurology Department, Athens Naval Hospital, 11521 Athens, Greece; doskastr@gmail.com (T.K.D.); kanspiliopoulos@hotmail.com (K.C.S.)
2   Neurology Department, Democritus University of Thrace, 68100 Alexandroupolis, Greece; christidi.f.a@gmail.com (F.C.); anniw_3@hotmail.com (A.T.); iiliop@hotmail.com (I.I.); vadikosm@yahoo.com (K.V.)
3   Neurology Department, University of Patras, 26504 Patras, Greece
4   Neurology Department, University of Cyprus, 20537 Nicosia, Cyprus; dantevavougios@hotmail.com
5   Psychiatry Department, Democritus University of Thrace, 68100 Alexandroupolis, Greece; tvorvola@med.duth.gr
6   Department of Physical Education and Sport Science, Democritus University of Thrace, 69100 Komotini, Greece; ckokkoti@affil.duth.gr (C.K.); nagelous@phyed.duth.gr (N.A.)
*   Correspondence: tsiptsios.dimitrios@yahoo.gr; Tel.: +30-694-432-0016

**Abstract:** Multiple sclerosis (MS) is a chronic immune-mediated disease of the central nervous system (CNS), characterized by the diffuse grey and white matter damage. Cognitive impairment (CI) is a frequent clinical feature in patients with MS (PwMS) that can be prevalent even in early disease stages, affecting the physical activity and active social participation of PwMS. Limited information is available regarding the influence of MS in social cognition (SC), which may occur independently from the overall neurocognitive dysfunction. In addition, the available information regarding the factors that influence SC in PwMS is limited, e.g., factors such as a patient's physical disability, different cognitive phenotypes, mood status, fatigue. Considering that SC is an important domain of CI in MS and may contribute to subjects' social participation and quality of life, we herein conceptualize and present the methodological design of a cross-sectional study in 100 PwMS of different disease subtypes. The study aims (a) to characterize SC impairment in PwMS in the Greek population and (b) to unveil the relationship between clinical symptoms, phenotypes of CI, mood status and fatigue in PwMS and the potential underlying impairment on tasks of SC.

**Keywords:** multiple sclerosis; social cognition; mood; fatigue; cognitive impairment

## 1. Introduction

Multiple sclerosis (MS) is a chronic immune-mediated disease of the central nervous system (CNS), characterized by diffuse grey and white matter damage. Patients with MS (PwMS) are presented with various phenotypes of functional disability, i.e., physical disability (motor weakness, spasticity, sensory dysfunction, visual loss, ataxia etc.), fatigue, pain, incontinence, cognitive impairment (CI), as well as psychosocial and behavioral dysfunction.

CI is common in PwMS even in early disease stages [1,2], affecting the physical activity and active social participation of PwMS [3]. Cognitive deficits more frequently pertain to attention, information processing speed, memory and executive function, having significant consequences for patients' daily living [4] beyond the ones associated with their physical symptoms [5,6]. CI may affect a person's ability to plan and execute physical

activities, leading to reduced engagement in exercise or other physical tasks, and can further hinder effective communication and interaction with others. More specifically, impaired attention and reduced processing speed can result in reduced motivation for physical activity due to difficulties in following instructions and coordinating movements, while executive impairment may have a negative effect on the adaptation to changing environments. Memory problems can lead to difficulties remembering conversations or appointments, resulting in challenges engaging in group conversations and potentially affecting interpersonal relationships. This can lead to social isolation and withdrawal due to the frustration associated with social interactions.

Social cognition (SC) has been recently examined in MS and found to be impaired (see recent meta-analyses [7–9]). SC is a multidimensional construct referring to a wide range of cognitive processes that allow humans to understand themselves and interact with and understand others, thus enabling them to form interpersonal relationships and engage in appropriate goal-directed behaviors [10]. These cognitive processes involve processing demands of emotional expressions (e.g., face, voice, body posture) as well as higher-order SC abilities, such as theory of mind (ToM) (i.e., making inferences about other subjects' mental states), making moral decisions, controlling emotions and feelings, and experiencing and expressing empathy [10,11]. Emotional processing relies on a range of cognitive abilities that work together to comprehend and navigate emotions. This involves recognizing and labeling emotions in oneself and others, understanding their underlying causes and effects, and empathizing with different perspectives. Effective regulation of emotions requires managing one's own feelings, while accurately expressing emotions through communication and body language. The ability to grasp the concept of others' thoughts and emotions, memory of emotional experiences, focused attention on emotional cues, and problem-solving skills all contribute to the intricate process. Anchored in self-awareness, emotional processing also demands adaptability, creativity, and a grasp of social dynamics. While autism spectrum disorder has been considered the archetypical disorder of SC [12], impaired SC abilities have been found in different neurodegenerative [13] and psychiatric diseases [14]. Emotion recognition and ToM are two vital processes since they are necessary for effective social interaction, enabling people to understand profound and subtle social cues [15] and allowing them to form and maintain relationships with others [16].

The relationships between dimensions of SC are not well understood, leaving unanswered the question as to whether these dimensions (e.g., emotional recognition and ToM) are independent or share underlying processes, thus a disruption of one may or may not lead to or predict a disruption in another. For example, previous research in typically developing children found that only perception of intention based on gaze, but not facial emotion recognition, was correlated with ToM [17]. Facial emotion recognition performance is worse in PwMS compared to healthy individuals [18–30], with significant differences observed for anger, sadness and fear, but not for disgust, happiness or surprise [8]. It has been suggested that the observed discrepancies between negative and positive emotions could be the result of PwMS' low sensitivity toward aversive stimuli [31], or a compensation mechanism that makes the processing of positive emotions easier than the processing of negative ones [24,32]. ToM performance is also worse in PwMS compared to healthy individuals [18,22,23,28,33–39]. Of note, considering the most commonly used measures of ToM, i.e., Faces test [40], Reading the Mind in the Eyes test [41], Faux Pas test [42], it appears that PwMS perform significantly worse on the Reading the Mind in the Eyes test and Faces Test but not on the Faux Pas test [33,35]. Methodological aspects related to the Faux Pas test (e.g., modification during different translations, use of child versions in adult populations), as well as the involvement of affective ToM components and emotion recognition in the Face Test and the Reading the Mind in the Eyes test compared to the cognitive ToM component in the Faux Pas test, may explain the previous discrepancies [8]. Thus, future studies are warranted to further examine these discrepancies.

Demographic characteristics and several MS-related features and symptoms including patients' functional disabilities, disease duration, MS subtypes, CI, depression, and fatigue, may be considered as important potential confounders of impaired SC in MS. A recent meta-analysis reported greater deficits in overall facial emotion recognition in older aged PwMS and a trend association with EDSS score and disease duration [8]. Indeed, there are studies indicating an association between EDSS score and facial emotion recognition [21,24,30] or ToM [39], while others do not detect such an association [23,25,34]. Progression rate has been found to be associated with ToM [35] and emotion recognition impairment [18,19,27,29]. In fact, impaired SC has been reported in relapsing-remitting MS (RRMS) [20,37,38,43] and primary progressive MS (PPMS) [44], as well as secondary progressive MS (SPMS) [20,30]. The direct comparison between RRMS and patients with PPMS [44] or chronic progredient MS [45] indicates better performance in RRMS. The severity of impaired SC is greater in PwMS with CI and correlates with the degree of CI [7–9,46]. Performance in SC tasks is associated with performance in neuropsychological tests of attention, processing speed, working memory, learning, and executive functions [21,22,24,25,29,34,36–39,47,48]. Nevertheless, the strength and the statistical significance of the reported associations is inconsistent [22,25,38]. In addition, it is unclear if SC impairment is secondary to the development of CI or whether it is an independent impairment.

Depression and fatigue are two another important MS-related symptoms. Of note, MS is a multidimensional disease and comorbid with psychiatric disorders, especially depression [49,50]. For example, depression is a common clinical feature in MS with a lifetime prevalence of approximately 50%. Furthermore, anxiety disorders may more frequently affect PwMS; yet, such reports are currently limited and less rigorous compared to studies investigating depression [51]. Fatigue is also among the common symptoms in MS, reported by approximately 75% of patients at some point during the disease [52] and often persisted even after improvement of cognitive performance and QoL following treatment [53]. Both depression and anxiety could affect the performance in cognitive tests [54,55], including SC. There is limited evidence on the relationship between impaired SC and higher rates of psychosocial fatigue, depressive symptoms, and anxiety levels in MS [56], even though other studies failed to identify an association between fatigue and SC performance in PwMS [57]. On the other hand, it may be possible that deficits of SC contribute to the presence of depression; for example, alexithymia constitutes a valuable prognostic factor of depression in MS, indicating that deficits in emotional perception could be associated with depressive symptoms [58]. Hence, the relationship between SC and depression, anxiety and fatigue has not thoroughly been investigated in the MS population, so that nonsecure conclusion could be based on current evidence [56]. Of note, there are studies that did not find an association between fatigue and SC [23,24,38].

Considering the insufficient number of studies and the inconsistent findings [7–9], we can therefore conclude that SC impairments in PwMS require further in-depth research with regards to possible associations between SC performance and MS-related symptoms. Although impaired SC is less acknowledged than non-SC impairment in MS (e.g., processing speed, working memory, learning, executive functions), it could be recognized as an important domain of CI in MS, contributing to the prognosis of social participation and quality of life (QoL). These intrapersonal social skills as well as emotional empathy could critically influence the functional status of PwMS in their daily routine. In fact, impaired SC has been correlated with a decline in social and mental domains of the quality of life in PwMS, even after adjusting for disease severity and duration, age and neurocognitive performance [8]. Therefore, further research is necessary since the negative impact of impaired SC in PwMS can be tremendous, resulting in unemployment, increased divorce rate and dissolution of partnerships [2,59–61], higher rates of social anxiety [62] and altered social interactions [61,63,64], and thus affecting patients' QoL [65,66]. Further research is expected to provide insights of the SC association with the MS-related symptoms, the neurocognitive impairment and their neural basis [8], both in cross-sectional as well as

longitudinal designs. In the study by Neuhaus et al., SC decline was limited to the affective component of SC and the decline was independent from the overall cognitive performance, EDSS score, disease duration and depression [67]. Fatigue has been associated with the performance in SC, indicating an origin from common neural tracts [67]. In fact, both fatigue and SC impairments in MS may share several anatomical abnormalities in the "cortico-striato-thalamo-cortical loop" and specific white matter tracts, e.g., uncinate fasciculus, corpus callosum, inferior fronto-occipital fasciculus, and cingulum [68]. These findings highlight the need for future research to determine the impact of SC deficits on the intrapersonal relationships of PwMS, as well as to validate the incidence of SC deficits in larger cohorts of patients with progressive MS [68,69]. To date, the relatively limited research of SC in MS included cross-sectional studies of small sample size, which primarily investigated patients with RRMS and a lower status of physical disability.

The main aims of the present study are: (a) the characterization of SC impairment in PwMS in the Greek population and (b) the relationship between clinical symptoms, cognitive scores, mood status and fatigue in PwMS and the potential underlying impairment on tasks of SC.

## 2. Methods

### 2.1. Ethics and Dissemination

The current study involving human participants has been reviewed and approved by the Hospital Local Ethical Committee of the Athens Naval Hospital (6265/15 June 2020). It will be conducted in accordance with the Declaration of Helsinki. All participants will provide written informed consent before their participation in the study.

### 2.2. Study Participants

Our prospective study will include 100 PwMS, who are regularly monitored in the outpatient clinic of the Neurology Department of Athens Naval Hospital. We will screen as many consecutive subjects as necessary to reach the final sample of 100 PwMS. According to the inclusion/exclusion criteria of the study, we will present in detail the number of subjects not included for any specific reason or due to a refusal to participate. All patients will answer a questionnaire regarding their demographic data in a specifically designated place of our clinic under supervision by the researcher, who will provide all necessary information and clarifications to the study participants.

### 2.3. Inclusion/Exclusion Criteria

The inclusion criteria for PwMS will be the following: (1) confirmed MS diagnosis based on the most recent diagnostic criteria, published in 2017, (2) Greek national who can fluently speak the Greek language, (3) age > 17 years old, education > 3 years, (4) Expanded Disability Status Scale (EDSS $\leq$ 8, (5) patients able to attend study visits in the clinic, to follow the instructions by the researcher and to answer questionnaires, (6) absence of severe and uncorrected visual or hearing problems that prevent neuropsychological evaluation (e.g., uncorrected visual acuity, or hearing loss), verified upon clinical evaluation and/or ophthalmological/otolaryngological examination, and (7) absence of severe cognitive dysfunction based on an initial mental screening test and local normative data for the Mini-Mental State Examination (MMSE) [70]. The exclusion criteria are the following: (1) a disease relapse in at least 1 month prior to the study assessments, (2) relapse during pregnancy, (3) use of corticosteroids, (4) presence of other concomitant diseases that cause fatigue, such as hypothyroidism (not treated), asthma or allergic reactions, (5) signs or symptoms indicative of active underlying infections, (6) current or past drug use or abuse. Study patients will be classified in two age groups, based on median age of the total sample, and two disability groups, based on the EDSS level of disability, i.e., no-to-moderate disability (0–3.5) and equal or more than moderate disability (EDSS $\geq$ 4). The suggested ranges are preliminary and could slightly vary based on the final sample size and its demographic characteristics. It is estimated that two distinct groups of functional disability

based on EDSS will be formed for each age group, resulting in a total number of four patient subgroups.

*2.4. Research Material*

Research material will include certain questionnaires and evaluation scales of different dimensions, which will involve demographic and baseline MS-related symptoms in the introduction section. We will include self-reported scales and neuropsychological tests that have been used in MS research previously and have been validated in the Greek population.

2.4.1. Self-Reported Scales

**Expanded Disability Status Scale (EDSS):** EDSS evaluates 8 functional systems that are associated with the CNS (pyramidal, cerebellar, brainstem, sensory, visual, bowel and bladder, cerebral and other functions of the neural system). Each functional system is rated with a specific EDSS score. EDSS step ranges from 0 to 10 [71].

**Modified Fatigue Impact Scale (MFIS):** The MFIS was proposed by the Multiple Sclerosis Council for Clinical Guidelines in 1998 as the appropriate rating scale of fatigue in the MS population, since it assesses the impact of fatigue in daily activities of PwMS as patients themselves perceive this, during the period of the last month. This scale discriminates between physical, cognitive, and psychosocial fatigue and has good validity and reliability. Answers are rated from 0 to 4 (0 = never, 1 = rarely, 2 = sometimes, 3 = often, 4 = almost always). Total score ranges from 0 to 84 and higher scores indicate greater fatigue. Study participants state with their answers the fatigue they experienced in the previous 4 weeks. Three discrete domains are included: (a) 10 questions for the cognitive factor, (b) 9 questions for the physical and (c) 2 questions for the psychosocial factor. Questions focus on fatigue symptoms (i.e., muscle weakness, the need to rest more often and longer, clumsiness) [72]. A Greek validation study is available [73].

**Beck Depression Inventory (BDI-II):** The Beck Depression Inventory scale was initially developed by Aaron T. Beck for assessments of depression severity and is one of the most widely used psychometric tests. BDI-II was published in 1996 and is the most recent version of this questionnaire. This questionnaire consists of 21 items for the evaluation of the severity of depression in adults and adolescents aged > 13 years. For all the 21 items (each question corresponds to a certain symptom), the examined subjects are asked to indicate the most suitable statement based on how they felt during the previous 2 weeks. Each question includes 4 statements that range from 0 to 3, where "0" indicates absence of the symptom and "3" corresponds to presence of the symptom at a severe grade [74]. A Greek validation study is available [53].

**Hospital Anxiety and Depression Scale (HADS):** HADS will be utilized to evaluate the incidence of developing mental disorders in PwMS. HADS is a useful tool to assess the mental state of hospitalized patients, since both anxiety and depression, which are the most common mental disorders in these patients, can be evaluated. This scale consists of 14 questions: 7 questions regarding the evaluation of developing anxiety disorders and the remaining 7 for depression assessments. Each sub-question includes 4 available answers, which are rated from 0 to 3. The total score of each disorder can vary between 0 and 21. Pathological scores are defined as >11, whereas a score < 7 is considered normal. Scores between 8 and 10 are characterized as borderline cases [75]. A Greek validation study and normative data are available [76].

**Multiple Sclerosis Quality of Life (MSQOL-54):** MSQOL-54 will be used in order to define the QoL in the study subjects [77]. This questionnaire utilizes the Short Form 36 (SF-36)-Item Health Survey questionnaire as the generic component, which is applied in assessments of QoL both in subjects with or without health problems. Eighteen items are also added to SF-36 that refer exclusively to PwMS. Thus, the complete questionnaire consists of 54 items that are classified in 12 subscales and 2 single-item questions. A Greek validation study is available [78].

2.4.2. Tests of Cognitive Functions

For study purposes, six (6) tests will be performed for assessments of the three (3) core cognitive functions that decline in MS, i.e., executive functions, memory and information processing speed. Considering that all cognitive measures are standardized in the local speaking population, standardized values will be obtained and used for further analysis, reflecting the degree of CI based on z-scores.

**Brief International Cognitive Assessment for Multiple Sclerosis (BICAMS) Battery:** The BICAMS will be used for the assessment of the following cognitive functions [79]. **(a)** *Information Processing Speed*: The Symbol Digit Modalities Test (SDMT) of the BI-CAMS battery constitutes a measure of information processing speed. It consists of boxes with numbers where each number corresponds to a specific symbol. Study subjects must assign, in a short time frame (90 s), a series of numbers to the corresponding symbols. Each correct answer receives 1 point. **(b)** *Verbal Memory*: Short- and long-term verbal memory will be evaluated via the California Verbal Learning Test-II (CVLT-II) of the BICAMS battery. Participants will have to memorize as many words as possible from a list of 16 nouns, which is read by the examiner over 5 learning trials. After each trial, subjects are asked to recall (short delay recall). After 30 min, subjects will be asked again to recall as many words as possible from the list (long delay recall). Each correct recall receives 1 point. The maximum score is 16. **(c)** *Visuospatial Memory*: In Brief Visuospatial Memory Test-Revised (BVMT-R) assessments, subjects are asked to see and memorize a page with 6 geometric figures in 10 s. Subsequently, subjects are asked to draw as many of the figures as possible. This process is repeated in 3 trials. Each figure receives 2 points if it is drawn accurately and in the correct position of the page, 1 point if one of the two requirements is met and 0 points in case of an absent recall. The total score is formed for the subscores of the three trials. A Greek validation study and normative data are available [80,81].

**Executive functions—Cognitive inhibition**: The Stroop Neuropsychological Screening Test (SNST) is comprised of two parts [82]. In the first, subjects are asked to read a list of color names as fast as they can (in a limit of 120 s). In the second part, which measures cognitive inhibition, subjects are instructed to ignore the color name and to name the color of the font as fast as they can (again up to 120 s). The score of cognitive inhibition is the sum of the responses (total read stimuli—stimuli where subject read the word instead of the font color) in the 120 s of the second part. A Greek validation study and normative data are available [83].

**Executive functions—Working memory**: A backward digital span will be used from the Wechsler Adult Intelligence Scale-IV [84]. Subjects will be instructed to repeat a list of numbers, backwards, with a graded difficulty. The most difficult task includes 8 digits. The final score equals the total sum of correct recalls and ranges between 0 and 16. For a more comprehensive assessment of the memory of numbers, a towards recall will be also included, as well as other tests from the Greek version of the WAIS scale. A Greek validation study and normative data are available (https://www.motiboaxiologisi.gr/tools.php?nid=2, accessed on 15 August 2023).

**Executive functions—Cognitive flexibility**: In the Trail Making Test [85], subjects are instructed to connect dots in order while alternating numbers and letters (1-A-2-B-3-C etc.). In the first part (A), subjects connect only numbers. In the second part (B), subjects connect both letters and numbers. The performance scoring is based on the time consumed to finish the test, with less time defining better scores. A Greek validation study and normative data are available [86,87].

**Social Cognition Tests—*"Reading the Mind in the Eyes" scale*:** This test consists of a set of 36 figures showing eye expressions, for which subjects should choose the available option that best describes what they feel or think about the figure. If more than one option seems compatible, subjects are instructed to choose only the most appropriate one of them. In this test, each correct answer receives 1 point, with a total score of 36 [41]. A Greek validation study and normative data of the adult population are available [88].

**Social Cognition Tests—*Faux Pas test*:** The affective and cognitive ToM will be assessed using the Faux Pas test, i.e., "something that should not be said". This test consists of 10 short stories, 5 scenarios containing a faux pas and 5 control stories. Faux pas scenarios include a situation, where a person unintentionally commits an insulting, unpleasant or impolite statement for another person. During the test, subjects are instructed to investigate the affective and cognitive situations to detect the faux pas in each scenario. Control scenarios are scored with 2 points if subjects do not detect faux pas, or with 0 points if subjects detect any faux pas. Total score ranges from 0 to 10. For each faux pas story, the score ranges from 0 to 6, resulting in a total of maximum 30 points. There is only 1 question in the test regarding cognitive ToM, so the total score for cognitive ToM ranges from 0 to 5. Accordingly, the 2 questions for the affective ToM indicate a total score ranging from 0 to 10. The maximum total score of the test is 32 [89,90]. A Greek validation study and normative data are available (https://www.autismresearchcentre.com/tests/faux-pas-test-adult/, accessed on 15 August 2023).

*2.5. Statistical Analysis*

Descriptive and inferential statistical analysis will be performed. We will consider internal consistencies for each rating scale, i.e., Cronbach's $\alpha$ between 0.50 and 0.95 ($0.50 < \alpha < 0.95$). The domains of the utilized neuropsychological tools and scales will be analyzed using the Categorical Principal Component Analysis, with the hypothesis of $n > 1$ dimensions and Cronbach's $\alpha$ between 0.50 and 0.95 ($0.50 < \alpha < 0.95$). Normality of the examined variables will be tested using Shapiro–Wilk test. To test the effect of age and functional disability (EDSS) on SC performance, ANOVA and Kruskal–Wallis tests, with post hoc analysis, will be performed, evaluating both main effects of age and functional disability, as well as interactions. To test the effect of MS subtype on SC performance, ANOVA and Kruskal-Wallis tests, with post hoc analysis, will be performed. To determine the factors affecting performance on SC tests in PwMS, two multiple regression analyses will be performed to identify significant predictors of SC performance on two SC tasks. SC scores will be entered as the dependent variable, and demographic data (age, gender, education), disease duration, EDSS score, cognition, mood and fatigue scores as independent variables without ordering. Furthermore, and as a secondary analysis for the QoL measurement, a multiple linear regression model will be conducted with MS-QOL-54 total score as the dependent variable, and demographic data, disease duration, EDSS score, cognition, SC, mood and fatigue scores as independent variables without ordering. Following previous studies [91], regression analyses will be conducted for different MS subtypes, considering the heterogeneity of clinical features between different MS subtypes. Statistics will be performed using SPSS v22.0. Statistical significance will be set at a $p$-value $< 0.05$, with Bonferroni correction for multiple comparisons in case of post hoc analyses.

**3. Discussion**

The present study protocol describes the design of a research project aiming to identify the profile of SC impairment in Greek adult PwMS and associations with patients' demographic, clinical and neurocognitive features, as well as mood status and fatigue.

Previous studies have shown that SC can be impaired in PwMS and can be often unrelated to other neurocognitive deficits [7], which has been also found in other neurodegenerative conditions [92]. The exact association between patients' demographic, clinical and neurocognitive features, as well as mood changes and fatigue and SC impairment has not yet been thoroughly examined. The assessment of SC impairment will be carried out using widely-used neuropsychological measures for the evaluation of SC in neurodegenerative conditions in general and MS specifically [7,13]. In addition, widely used and local population-validated and standardized clinical scales, neuropsychological measures and psychometric questionnaires will be administered to evaluate patients' clinical profile, neurocognitive status, mood changes and fatigue.

The present study is not without limitations. One of the major limitations is the cross-sectional design of the study which will not allow certain conclusions regarding causal directions between SC and previous-reported factors that can be related to SC impairment. Thus, future longitudinal studies are warranted to overcome this limitation.

Despite possible limitations, the present study represents the first national attempt to thoroughly examine the profile of SC in Greek PwMS and factors associated with SC impairment.

**Author Contributions:** T.K.D., T.V., I.I. and K.V. compiled the protocol. F.C., K.C.S., D.T., G.D.V. and A.T. structured research material. N.A. and C.K. structured statistical analysis. The corrected version was discussed collegially. T.K.D. and F.C. wrote the final version. All authors have read and agreed to the published version of the manuscript.

**Funding:** This work was supported by the project "Study of the interrelationships between neuroimaging, neurophysiological and biomechanical biomarkers in stroke rehabilitation (NEURO-BIOMECH in stroke rehab)" (MIS 5047286), which is implemented under the action of "Support for Regional Excellence", funded by the operational program "Competitiveness, Entrepreneurship and Innovation" (NSRFm2014-2020) and co-financed by Greece and the European Union (the European Regional Development Fund).

**Institutional Review Board Statement:** The study was conducted in accordance with the Declaration of Helsinki, and approved by the Institutional Review Board (or Ethics Committee) of The Athens Naval Hospital (protocol code: 6265, date of approval: 15 June 2020).

**Informed Consent Statement:** Informed consent was obtained from all subjects involved in the study.

**Data Availability Statement:** All data discussed within this manuscript are available on PubMed.

**Conflicts of Interest:** The authors declare no conflict of interest.

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
