# Peer review of "Social Cognition Impairments in Association to Clinical, Cognitive, Mood, and Fatigue Features in Multiple Sclerosis: A Study Protocol"

_2035-8377, doi:10.3390/neurolint15030068_

Round 1

Reviewer 1 Report

Doskas and colleagues present their protocol for a cross-sectional study of 100 MS patients in Greece that aims to characterize social cognitive impairment and elucidate the relationship between clinical characteristics, MS courses, mood and fatigue. The introduction requires significant reworking per my targeted feedback below.

Main feedback:

General:

-        The background was challenging to follow and would benefit from significant reorganization and a more comprehensive review of the existing literature. In particular, describing the interrelationship between emotion recognition and higher-order social cognition abilities like ToM and what has been found in the MS literature regarding emotion recognition and ToM. The authors are encouraged to discuss what gaps the work will address (i.e., that few studies have considered the potential association between SC and clinical factors).

-      For clarity, I would recommend being consistent in how you describe what your work is looking at (e.g., always use MS-related symptoms OR MS characteristics)

-       I would also be cautious about the use of “social cognitive disorder” and instead refer to challenges in the domain or on tasks of social cognition

Abstract:

-     Rephrase lines 23-24 for clarity and brevity

Introduction:

-          Line 36 - Can you clarify what is meant by “disability associated with the executive function”?

-          Line 38 – take out “clinical feature” and state that CI is common in PwMS even in early disease stages and specify how it affects physical activity and social engagement;  more references are needed here

-          Line 44 – information is limited but a brief review of what is known about CI and HRQOL in MS is warranted (e.g., association or lackthereof between CI and mood, fatigue, etc.)

-          Line 48 – “…it also includes distinguishable but overlapping cognitive and emotional components.” Such as? It may be clearer to describe the cognitive abilities required for emotional processing (e.g., attention, processing speed, etc.)

-          Lines 58-62 – unsure what the first sentence in this paragraph is getting at, particularly as on line 67 you mention that SC deficit severity has been associated with CI (this contradicts the line that says that SC impairment may occur independently from overall neurocog dysfunction).   I am also unclear why ADHD and depression are specifically mentioned – would consider reworking or deleting these lines (line 61)

-          Line 84 – component vs. compound?

-          What neural tracts have been associated with both fatigue and SC? Would specify here.

-          Lines 87-89 are difficult to comprehend, it is unclear why EF is mentioned in a sentence with disability and mood. Perhaps more context is needed.

-          The justification for the study as it is presented on lines 89-91 does not seem to relate to the study findings presented in the brief overview of the literature. As noted above, more detailed literature review is necessary to highlight the gap and how your work will address this.

Please define what is meant by phenotypes of CI (line 113)

Methods:

-          Inclusion/exclusion criteria:

o   How will you assess for severe cognitive dysfunction to identify participants for exclusion?  

o   Will functional disability be assessed with EDSS? Please clarify.

o   How were age bands determined and for what purpose?

-          Measures:

o   If the measure has been validated/used in MS research previously, I would state this (for each measure)

-          Statistical analysis:

o   Given this is a protocol, I would encourage the authors to be more detailed in the analyses that will be used to address each aim. Computing descriptive statistics to characterize the sample is fine, but I would like to see how analyses will address specific hypotheses related to the second aim of testing the relationship between clinical characteristics and everything else. 

o   Clarify what you mean by qualitative variables and why chi-square is the appropriate statistical test for this/how this will be used for continuous outcome variables. Alternately specify which dichotomous variables the chi-square will be used for.

o   How will you control or consider heterogeneous MS subtypes? This should be explicitly stated.

o   If you plan to compare outcomes across age groups, how will you control for this? It is possible that you are not interested in this and will only be comparing patients with “functional disability” to those without in each age band, perhaps mention this outright.

o   What is the rationale for correlation vs. multiple regression? This could be a lot of outcomes for correlational analyses.

Please see above. Some editing is required to clarify some points and remove redundant information.

Author Response

Dear Reviewer,

Many thanks for your time spent reviewing our manuscript.

According to your comments, appropriate modifications were made to the text as follows:

Main feedback:

General:

-        The background was challenging to follow and would benefit from significant reorganization and a more comprehensive review of the existing literature. In particular, describing the interrelationship between emotion recognition and higher-order social cognition abilities like ToM and what has been found in the MS literature regarding emotion recognition and ToM. The authors are encouraged to discuss what gaps the work will address (i.e., that few studies have considered the potential association between SC and clinical factors).

Response: We thank the reviewer for dedicating time to review our study protocol and providing such constructive suggestions. According to the detail feedback provided by the reviewer, we made significant changes to the Introduction section and clarify all necessary points.

“Multiple sclerosis (MS) is a chronic immune-mediated disease of the central nervous system (CNS), characterized by diffuse grey and white matter damage. Patients with MS (PwMS) are presented with various phenotypes of functional disability, i.e. physical disability (motor weakness, spasticity, sensory dysfunction, visual loss, ataxia etc.), fatigue, pain, incontinence, cognitive impairment (CI), as well as psychosocial and behavioral dysfunction.

CI is common in PwMS even in early disease stages [1,2], affecting the physical activity and active social participation of PwMS [3]. Cognitive deficits more frequently pertain to attention, information processing speed, memory and executive function , having significant consequences for patients’ daily living [4] beyond the ones associated with their physical symptoms [5,6]. CI may affect a person's ability to plan and execute physical activities, leading to reduced engagement in exercise or other physical tasks, and can further hinder effective communication and interaction with others. More specifically, impaired attention and reduced processing speed can result in reduced motivation for physical activity due to difficulties in following instructions and coordinating movements, while executive impairment may have a negative effect on the adaptation to changing environments. Memory problems can lead to difficulties remembering conversations or appointments, resulting in challenges engaging in group conversations and potentially affecting interpersonal relationships. This can lead to social isolation and withdrawal due to the frustration associated with social interactions.

Social cognition (SC) has relatively been recently examined in MS and found to be impaired (see recent meta-analyses [7–9]). SC is a multidimensional construct referring to a wide range of cognitive processes that allow humans to understand themselves and interact with and understand others, thus enabling them to form interpersonal relationships and engage in appropriate goal-directed behaviors [10]. These cognitive process involve processing demands of emotional expressions (e.g. face, voice, body posture) as well as higher-order SC abilities, such as theory of mind [ToM] (i.e. making inferences about other subjects’ mental states), making moral decisions, controlling emotions and feelings, and experiencing and expressing empathy [10,11]. Emotional processing relies on a range of cognitive abilities that work together to comprehend and navigate emotions. This involves recognizing and labeling emotions in oneself and others, understanding their underlying causes and effects, and empathizing with different perspectives. Effective regulation of emotions requires managing one's own feelings, while accurately expressing emotions through communication and body language. The ability to grasp the concept of others' thoughts and emotions, memory of emotional experiences, focused attention on emotional cues, and problem-solving skills all contribute to the intricate process. Anchored in self-awareness, emotional processing also demands adaptability, creativity, and a grasp of social dynamics. With autism spectrum disorder been the archetypical disorder of SC [12], impaired SC abilities have been found in different neurodegenerative [13] and psychiatric diseases [14]. Emotion recognition and ToM are two vital processes since they are necessary for effective social interaction, enabling people to understand profound and subtle social cues [15] and allowing them to form and maintain relationships with others [16].

The relationships between dimensions of SC are not well understood, leaving unanswered the question whether these dimensions (e.g. emotional recognition and ToM) are independent or share underlying processes, thus a disruption of one may or may not lead to or predict a disruption in another. For example, previous research in typically developing children found that only perception of intention based on gaze but not facial emotion recognition was correlated with ToM [17]. Facial emotion recognition performance is worse in PwMS compared to healthy individuals [18–30], with significant differences been observed for anger, sadness and fear but not for disgust, happiness or surprise [8]. It has been suggested that the observed discrepancies between negative and positive emotions could be the result of PwMS low sensitivity toward aversive stimuli [31] or a compensation mechanism which makes the process of positive emotions easier than the process of negative ones [24,32]. ToM performance is also worse in PwMS compared to healthy individuals [18,22,23,28,33–39]. Of note, considering the most commonly used measures of ToM, i.e. Faces test [40], Reading the Mind in the Eyes test [41], Faux Pas test [42], it appears that PwMS performed significantly worse on the Reading the Mind in the Eyes test and Faces Test but not on Faux Pas test [33,35]. Methodological aspects related to the Faux Pas test (e.g. modification during different translations, use of child versions in adult populations) as well as the involvement of affective ToM component and emotion recognition in Face Test and Reading the Mind in the Eyes test compared to the cognitive ToM component in the Faux Pas test may explain the previous discrepancies [8]. Thus, future studies are warranted to further examine these discrepancies.

Demographic characteristics and several MS-related features and symptoms including patients’ functional disability, disease duration, MS subtypes, CI, depression, and fatigue, may be considered as important potential confounders of impaired SC in MS. A recent meta-analysis reported greater deficits in overall facial emotion recognition in older aged PwMS and a trend association with EDSS score and disease duration [8]. Indeed, there are studies indicating an association between EDSS score and facial emotion recognition [21,24,30] or ToM [39] while others not detecting such an association [23,25,34]. Progression rate has been found to be associated with ToM [35] and emotion recognition impairment [18,19,27,29]. In fact, impaired SC has been reported in relapsing-remitting MS (RRMS) [20,37,38,43], primary progressive MS (PPMS) [44], as well as secondary progressive MS (SPMS)  [20,30]. The direct comparison between RRMS and patients with PPMS [44] or chronic progredient MS [45] indicates better performance in RRMS. The severity of impaired SC is greater in PwMS with CI and correlates with the degree of CI [7–9,46]. Performance in SC tasks is associated with performance in neuropsychological tests of attention, processing speed, working memory, learning, and executive functions [21,22,24,25,29,34,36–39,47,48]. Nevertheless, the strength and the statistical significance of the reported associations is inconsistent [22,25,38]. In addition, it is unclear if SC impairment is secondary to the development of CI or whether it is an independent impairment.

Depression and fatigue are two another important MS-related symptoms. Of note, MS is a multidimensional disease and comorbid with psychiatric disorders, especially depression [49,50]. For example, depression a common clinical feature in MS with a lifetime prevalence of approximately 50%. Furthermore, anxiety disorders may more frequently affect PwMS; yet, such reports are currently limited and less rigorous as compared to studies investigating depression [51]. Fatigue is also among the common symptoms in MS, reported by approximately 75% of patients at some point during the disease [52] and often persisted even after improvement of cognitive performance and QoL following treatment [53]. Both depression and anxiety could affect the performance in cognitive tests [54,55], including SC. There is limited evidence on the relationship between impaired SC and higher rates of psychosocial fatigue, depressive symptoms, and anxiety levels in MS [56], even though other studies failed to identify an association between fatigue and SC performance in PwMS [57]. On the other hand, it may be possible that deficits of SC contribute to the presence of depression; for example, alexithymia constitutes a valuable prognostic factor of depression in MS, indicating that deficits in emotional perception could be associated with depressive symptoms [58]. Hence, the relationship between SC and depression, anxiety and fatigue has not thoroughly been investigated in MS population, so that nonsecure conclusion could be based on current evidence [56]. Of note, there are studies that did not find an association between fatigue and SC [23,24,38].

Considering the insufficient number of studies and the inconsistent findings [7–9], we can therefore conclude that SC impairments in PwMS require further in-depth research with regards to possible associations between SC performance and MS-related symptoms. Although impaired SC is less acknowledged than non-SC impairment in MS (e.g. processing speed, working memory, learning, executive functions), it could be recognized as an important domain of CI in MS, contributing to the prognosis of social participation and quality of life (QoL). These intrapersonal social skills as well as emotional empathy could critically influence the functional status of PwMS in their daily routine. In fact, impaired SC has been correlated with a decline in social and mental domains of the quality of life in PwMS, even after adjusting for disease severity and duration, age and neurocognitive performance [8]. Therefore, further research is necessary since the negative impact of impaired SC in PwMS can be tremendous, resulting in unemployment, increased divorce rate and dissolution of partnerships [2,59–61], higher rates of social anxiety [62] and altered social interactions [61,63,64], and thus affecting patients’ QoL [65,66]. Further research is expected to provide insights of the SC association with the MS-related symptoms, the neurocognitive impairment and their neural basis [8], both in cross-sectional as well as longitudinal designs. In the study by Neuhaus et al., SC decline was limited to the affective component of SC and the decline was independent from the overall cognitive performance, EDSS score, disease duration and depression [67]. Fatigue has been associated with the performance in SC, indicating an origin from common neural tracts [67]. In fact both fatigue and SC impairments in MS may share several anatomical abnormalities in the “cortico-striato-thalamo-cortical loop” and specific white matter tracts, e.g. uncinate fasciculus, corpus callosum, inferior fronto-occipital fasciculus, and cingulum [68]. These findings highlight the need for future research to determine the impact of SC deficits on the intrapersonal relationships of PwMS, as well as to validate the incidence of SC deficits in larger cohorts of patients with progressive MS [68,69]. Up to date, the relatively limited research of SC in MS included cross-sectional studies of small sample size, which primarily investigated patients with RRMS and lower status of physical disability.

The main aims of the present study are: (a) the characterization of SC impairment in PwMS of Greek population and (b) the relationship between clinical symptoms, cognitive scores, mood status and fatigue in PwMS and the potential underlying impairment on tasks of SC.

-      For clarity, I would recommend being consistent in how you describe what your work is looking at (e.g., always use MS-related symptoms OR MS characteristics)

Response: We thank the reviewer for the suggestion. We made all necessary changes and now use the word symptoms throughout the text.

-       I would also be cautious about the use of “social cognitive disorder” and instead refer to challenges in the domain or on tasks of social cognition

Response: Thank you for the suggestion. Instead of using “social cognitive disorder” we now use “impairment on tasks of SC”.

Abstract:

-     Rephrase lines 23-24 for clarity and brevity

Response: We thank the reviewer for highlighting this point. We rephrased lines 23-24 and split the original sentence into two sentences which are now read as follows: “Considering that SC is an important domain of CI in MS and may contribute to subjects’ social participation and quality of life, we herein conceptualize and present the methodological design of a cross-sectional study in 100 PwMS of different disease subtypes. The study aims (a) to characterize SC impairment in PwMS of Greek population and (b) to unveil the relationship between clinical symptoms, phenotypes of CI, mood status and fatigue in PwMS and the potential underlying impairment on tasks of SC.”

Introduction:

-          Line 36 - Can you clarify what is meant by “disability associated with the executive function”?

Response: Thank you for the comment. We rephrased it and it is now read as “cognitive impairment (CI)".

-          Line 38 – take out “clinical feature” and state that CI is common in PwMS even in early disease stages and specify how it affects physical activity and social engagement; more references are needed here

Response: We really appreciate the reviewer’s comment. We have re-written the paragraph according to the suggestions, also providing additional references.CI is common in PwMS even in early disease stages [1,2], affecting the physical activity and active social participation of PwMS [3]. Cognitive deficits more frequently pertain to attention, information processing speed, memory and executive function , having significant consequences for patients’ daily living [4] beyond the ones associated with their physical symptoms [5,6]. CI may affect a person's ability to plan and execute physical activities, leading to reduced engagement in exercise or other physical tasks, and can further hinder effective communication and interaction with others. More specifically, impaired attention and reduced processing speed can result in reduced motivation for physical activity due to difficulties in following instructions and coordinating movements, while executive impairment may have a negative effect on the adaptation to changing environments. Memory problems can lead to difficulties remembering conversations or appointments, resulting in challenges engaging in group conversations and potentially affecting interpersonal relationships. This can lead to social isolation and withdrawal due to the frustration associated with social interactions.”

-          Line 44 – information is limited but a brief review of what is known about CI and HRQOL in MS is warranted (e.g., association or lack thereof between CI and mood, fatigue, etc.)

Response: Thank you for the comment. Based on the reviewer’s previous comment for the background and the link between the presented studies and the aim of the study, we deleted this paragraph since it did not provide relevant information.

-          Line 48 – “…it also includes distinguishable but overlapping cognitive and emotional components.” Such as? It may be clearer to describe the cognitive abilities required for emotional processing (e.g., attention, processing speed, etc.)

Response: We thank the reviewer for the suggestion. The following sentences are now included in the revised version of the manuscript, Emotional processing relies on a range of cognitive abilities that work together to comprehend and navigate emotions. This involves recognizing and labeling emotions in oneself and others, understanding their underlying causes and effects, and empathizing with different perspectives. Effective regulation of emotions requires managing one's own feelings, while accurately expressing emotions through communication and body language. The ability to grasp the concept of others' thoughts and emotions, memory of emotional experiences, focused attention on emotional cues, and problem-solving skills all contribute to the intricate process. Anchored in self-awareness, emotional processing also demands adaptability, creativity, and a grasp of social dynamics”.

-          Lines 58-62 – unsure what the first sentence in this paragraph is getting at, particularly as on line 67 you mention that SC deficit severity has been associated with CI (this contradicts the line that says that SC impairment may occur independently from overall neurocog dysfunction).   I am also unclear why ADHD and depression are specifically mentioned – would consider reworking or deleting these lines (line 61)

Response: We do acknowledge that these two sentences are contradictory and we do apologize for the phrasing. We rephrased the sentences and in the revised version of the manuscript, the information is now presented as follows, “The severity of impaired SC is greater in PwMS with CI and correlates with the degree of CI [7–9,46]. Performance in SC tasks is associated with performance in neuropsychological tests of attention, processing speed, working memory, learning, and executive functions [21,22,24,25,29,34,36–39,47,48]. Nevertheless, the strength and the statistical significance of the reported associations is inconsistent [22,25,38]. In addition, it is unclear if SC impairment is secondary to the development of CI or whether it is an independent impairment.” Regarding the reference to ADHD and depression, we would like to present different levels of impaired SC in different neurological and psychiatric diseases based on the available studies. Based on the reviewer’s comment, we carefully thought this point and we deleted the studies and any reference to ADHD and depression to avoid any misunderstanding. The previous sentence is now read as follows “With autism spectrum disorder been the archetypical disorder of SC [12], impaired SC abilities have been found in different neurodegenerative [13] and psychiatric diseases [14].”

-          Line 84 – component vs. compound?

Response: We thank the reviewer for identifying this error. We changed it to “component”.

-          What neural tracts have been associated with both fatigue and SC? Would specify here.

Response: Based on the reviewer’s suggestion, we have answered this question in the revised version of the Introduction, “In fact both fatigue and SC impairments in MS may share several anatomical abnormalities in the “cortico-striato-thalamo-cortical loop” and specific white matter tracts, e.g. uncinate fasciculus, corpus callosum, inferior fronto-occipital fasciculus, and cingulum [68].”

-          Lines 87-89 are difficult to comprehend, it is unclear why EF is mentioned in a sentence with disability and mood. Perhaps more context is needed.

Response: We thank the reviewer for the important highlight. We deleted this sentence in the revised version of the manuscript.

-          The justification for the study as it is presented on lines 89-91 does not seem to relate to the study findings presented in the brief overview of the literature. As noted above, more detailed literature review is necessary to highlight the gap and how your work will address this.

Response: We appreciate reviewer’s comments to increase the quality of the manuscript. According to the suggestions, we carefully re-organized the introduction and made all necessary changes adding representative and appropriate studies. We now believe that the aim of the study related to the study findings presented in the overview of the literature.

- Please define what is meant by phenotypes of CI (line 113)

Response: We thank the reviewer for letting us provide additional information regarding this point. Phenotypes of CI refer to cognitive dysfunction which is characterized by various patterns: decreased information processing speed, memory challenges, compromised attention and concentration, impaired executive functions, difficulties in verbal and visual processing, compromised spatial awareness, and language-related deficits. The severity and combination of these impairments vary among individuals, impacting their ability to manage daily tasks, work, and social interactions. However, according to further discussion within the group, we decided to use cognitive scores as continuous variables and not subgroups based on phenotypes of CI. Considering that all cognitive measures are standardized in local speaking population, standardized values will be obtained and used for further analysis, reflecting the degree of CI based on z-scores. This information is now included in the Methods section. Thank you!

Methods:

-          Inclusion/exclusion criteria:

o   How will you assess for severe cognitive dysfunction to identify participants for exclusion?  

Response: Thank you for the comment. We included this information on the revised version of the manuscript, 7) absence of severe cognitive dysfunction based on initial mental screening test and local normative data for the Mini-Mental State Examination (MMSE) [70]”.

o    Will functional disability be assessed with EDSS? Please clarify.

Response: We apologize for not clarifying this point. Functional disability will be assessed with EDSS and this is now reported in the Methods section, “… and two disability groups based on the EDSS level of disability, i.e. no-to-moderate disability (0-3.5) and equal or more than moderate disability (EDSS≥4).

o   How were age bands determined and for what purpose?

Response: The age bands were determined to evaluate any causative role of age in SC impairment in PwMS, as well as in the association between SC and MS-related symptoms, cognitive impairment, mood status and fatigue.  We do agree that the age bands were not determined on a formal basis. Based on the reviewer comment, we further discussed this point within the team, and we decided that the use of the sample median age would be more appropriate to determine two age groups (i.e. younger and older PwMS). We made all necessary changes to the revised version of the manuscript to clarify this issue, “Study patients will be classified in two age groups based on median age of the total sample and two disability groups based on the EDSS level of disability, i.e. no-to-moderate disability (0-3.5) and equal or more than moderate disability (EDSS≥4).” Thank you!

-          Measures:

o   If the measure has been validated/used in MS research previously, I would state this (for each measure)

Response: All the measures provided in the present study protocol have been extensively used in MS research previously. Considering that there are thousands of studies using these measures, it would be biased to cite only a couple of them. However, we state this at the beginning of the section 2.4. Research material, “We will include self-reported scales and neuropsychological tests that have been used in MS research previously and have been validated in Greek population.Thank you for the suggestion!

-          Statistical analysis:

o   Given this is a protocol, I would encourage the authors to be more detailed in the analyses that will be used to address each aim. Computing descriptive statistics to characterize the sample is fine, but I would like to see how analyses will address specific hypotheses related to the second aim of testing the relationship between clinical characteristics and everything else. 

Response: We thank the reviewer for the suggestion. We have revised the statistical analysis paragraph and elaborate on the statistical analysis regarding the second aim of the study.Descriptive and inferential statistical analysis will be performed. We will consider internal consistencies for each rating scale, i.e. Cronbach’s α between 0.50 and 0.95 (0.50<α<0.95). The domains of the utilized neuropsychological tools and scales will be analyzed using the Categorical Principal Component Analysis, with the hypothesis of n>1 dimensions and Cronbach’s α between 0.50 and 0.95 (0.50<α<0.95). Normality of the examined variables will be tested using Shapiro-Wilk test. To test the effect of age and functional disability (EDSS) on SC performance, ANOVA and Kruskal-Wallis tests, with post-hoc analysis, will be performed, evaluating both main effects of age and functional disability, as well as interactions. To test the effect of MS subtype on SC performance, ANOVA and Kruskal-Wallis tests, with post-hoc analysis, will be performed.   To determine the factors affecting performance on SC tests in PwMS, two multiple regression analyses will be performed to identify significant predictors of SC performance on two SC tasks. SC scores will be entered as dependent variable, and demographic data (age, gender, education), disease duration, EDSS score, cognition, mood and fatigue scores as independent variables without ordering. Furthermore and as a secondary analysis for the QoL measurement, multiple linear regression model will be conducted with MS-QOL-54 total score as dependent variable, and demographic data, disease duration, EDSS score, cognition, SC, mood and fatigue scores as independent variables without ordering. Following previous studies [92], regression analyses will be conducted for different MS subtypes considering the heterogeneity of clinical features between different MS subtypes. Statistics will be performed using SPSS v22.0. Statistical significance will be set at a p-value<0.05, with Bonferroni correction for multiple comparisons in case of post-hoc analyses.”  

o   Clarify what you mean by qualitative variables and why chi-square is the appropriate statistical test for this/how this will be used for continuous outcome variables. Alternately specify which dichotomous variables the chi-square will be used for.

Response: The sentence was deleted. Thank you.

o   How will you control or consider heterogeneous MS subtypes? This should be explicitly stated.

Response: We do acknowledge that heterogeneity in MS subtypes is one the major limitations of designing and implementing a study, as well as interpreting the results. The following step will be applied based on previous studies, “Following previous studies [92], regression analyses will be conducted for different MS subtypes considering the heterogeneity of clinical features between different MS subtypes.” We also acknowledge in advance that any interpretation of the findings should be expressed in caution and considering that the heterogeneity is an inherent limitation in such studies.

o   If you plan to compare outcomes across age groups, how will you control for this? It is possible that you are not interested in this and will only be comparing patients with “functional disability” to those without in each age band, perhaps mention this outright.

Response: We elaborated on this point and added the following sentence, “To test the effect of age and functional disability (EDSS) on SC performance, ANOVA and Kruskal-Wallis tests, with post-hoc analysis, will be performed, evaluating both main effects of age and functional disability, as well as interactions.”. Thank you!

o   What is the rationale for correlation vs. multiple regression? This could be a lot of outcomes for correlational analyses.

Response: We thank the reviewer for the suggestion. We have revised this part of the statistical analyses and we do agree that multiple regression analyses can address the aim of the study.

Comments on the Quality of English Language

Please see above. Some editing is required to clarify some points and remove redundant information.

Response: We really appreciate reviewer’s suggestions to clarify some points and remove redundant information.

Looking forward to your follow up comments.

Yours Sincerely,

Dr Tsiptsios

Reviewer 2 Report

In the present Protocol, the Authors aimed to evaluate (a) the characterization of social cognition impairment in patients with Multiple Sclerosis (PwMS) of the Greek population and (b) the relationship between clinical characteristics, phenotypes of cognitive impairment, mood status and fatigue in PwMS and the potential underlying SC disorder.

Overall, I found this protocol interesting, original and scientifically sound. However, I have several suggestions aimed at improving the quality of the paper, and these are outlined below:

1) In the introduction, a brief note on the fact that MS might be multidimensional and comorbid with psychiatric disorder, especially depression, should be more in-depth discussed with with appropriate references (please see and refer to the following dois: 10.1080/14656566.2018.1516207 and 10.1007/s40263-018-0489-5).

2) I suggest adding relevant studies concerning the protocol topic, perhaps condensed in a Table which would be very informative.

3) Please also consider how many subjects will be screened to reach the final sample, and how many will not be included for any reason (and why) or refuse to participate. Please, add some more informations on this point. 

4) Evaluated the presence of an eventual intellectual disability taken into consideration. As well the Authors excluded patients diagnosed with visual problems and severe cognitive dysfunction, how was such evaluation carried out?

5) As well was take into consideration the possible presence of current or past drug use or abuse as it is frequent in PwMS. 

6) Please, specify if internal consistencies for each rating scale will be calculated.

Author Response

Dear Reviewer,

Many thanks for your time spent reviewing our manuscript.

According to your comments, appropriate modifications were made to the text as follows:

In the present Protocol, the Authors aimed to evaluate (a) the characterization of social cognition impairment in patients with Multiple Sclerosis (PwMS) of the Greek population and (b) the relationship between clinical characteristics, phenotypes of cognitive impairment, mood status and fatigue in PwMS and the potential underlying SC disorder.

Overall, I found this protocol interesting, original and scientifically sound. However, I have several suggestions aimed at improving the quality of the paper, and these are outlined below:

Response: We thank the reviewer for dedicating time to review our study protocol and providing such encouraging comments.

1) In the introduction, a brief note on the fact that MS might be multidimensional and comorbid with psychiatric disorder, especially depression, should be more in-depth discussed with with appropriate references (please see and refer to the following dois: 10.1080/14656566.2018.1516207 and 10.1007/s40263-018-0489-5).

Response: We thank the reviewer for the suggestion. We highlight the multidimensionality of the disease and the comorbidity with psychiatric disorders, further including the studies suggested by the reviewer, “Of note, MS is a multidimensional disease and comorbid with psychiatric disorders, especially depression [49,50].”

2) I suggest adding relevant studies concerning the protocol topic, perhaps condensed in a Table which would be very informative.

Response: We appreciate reviewer’s comment. We thoroughly discussed this point within the team. In the revised version of the manuscript/introduction section, we have now included additional relevant studies concerning the protocol topic. In addition, considering that recent meta-analyses have presented similar information, we now reference these studies for further information. Thank you!

3) Please also consider how many subjects will be screened to reach the final sample, and how many will not be included for any reason (and why) or refuse to participate. Please, add some more information on this point.

Response: We will screen as many consecutive subjects as necessary to reach the final sample of 100 PwMS. According to the inclusion/exclusion criteria, we will finally present in detail the number of subjects not included for any specific reason or refuse to participate. We added this information in the revised version, Study Participants section, “We will screen as many consecutive subjects as necessary to reach the final sample of 100 PwMS. According to the inclusion/exclusion criteria of the study, we will finally present in detail the number of subjects not included for any specific reason or refuse to participate.” Thank you!

4) Evaluated the presence of an eventual intellectual disability taken into consideration. As well the Authors excluded patients diagnosed with visual problems and severe cognitive dysfunction, how was such evaluation carried out?

Response: We apologize for not clarifying this point. The following details are now provided, “(6) absence of severe and uncorrected visual or hearing problems that prevent neuropsychological evaluation (e.g. uncorrected visual acuity, or hearing loss), verified upon clinical evaluation and/or opthalmogical/otolaryngological examination, and (7) absence of severe cognitive dysfunction based on initial mental screening test and local normative data for the Mini-Mental State Examination (MMSE) [70]”.

5) As well was take into consideration the possible presence of current or past drug use or abuse as it is frequent in PwMS. 

Response: We really appreciate reviewer’s suggestion. We discussed this point within our team, and we decided to add it as an exclusion criterion on our study, “(6) current or past drug use or abuse”. Thank you!

6) Please, specify if internal consistencies for each rating scale will be calculated.

Response: We thank the reviewer for the question. We will consider internal consistencies for each rating scale, i.e. Cronbach’s α between 0.50 and 0.95 (0.50<α<0.95).

Looking forward to your follow up comments.

Yours Sincerely,

Dr Tsiptsios

Round 2

Reviewer 1 Report

The authors have done a good job revising the manuscript. They have provided relevant details to ensure the study constructs are well explained and the aim of the study is situated within the existing literature on the topic. 

One edit on page 2, line 52. Delete word "relatively"

One final suggestion is that the authors modify the title to state that this is a protocol? This wording might make it clearer to anyone who is doing a literature search on this topic. 

Fine

Author Response

Reviewer: The authors have done a good job revising the manuscript. They have provided relevant details to ensure the study constructs are well explained and the aim of the study is situated within the existing literature on the topic. 

Response: We thank the reviewer for dedicating time to review the revised version of f the manuscript and we really appreciate the encouraging comments.

Reviewer: One edit on page 2, line 52. Delete word "relatively"

Response: We deleted the word “relatively”, page 2, line 52. Thank you.

Reviewer:  One final suggestion is that the authors modify the title to state that this is a protocol? This wording might make it clearer to anyone who is doing a literature search on this topic. 

Response: According to the reviewer’s suggestion, we changed the title and included the phrase “a study protocol”. The title is now read as follows:  “Social cognition impairments in association to clinical, cognitive, mood, and fatigue features in multiple sclerosis: a study protocol”. Thank you.